# Amino acid sensor GCN2 promotes SARS-CoV-2 receptor ACE2 expression in response to amino acid deprivation

Xiaoming Hu[1,3], Yuguo Niu[1,2,3], Peixiang Luo[2], Fei Xiao[1], Feixiang Yuan[1], Hanrui Yin[2], Shanghai Chen[1] & Feifan Guo [1✉]

Angiotensin-converting enzyme 2 (ACE2) has been identified as a primary receptor for severe acute respiratory syndrome coronaviruses 2 (SARS-CoV-2). Here, we investigated the expression regulation of ACE2 in enterocytes under amino acid deprivation conditions. In this study, we found that ACE2 expression was upregulated upon all or single essential amino acid deprivation in human colonic epithelial CCD841 cells. Furthermore, we found that knockdown of general control nonderepressible 2 (GCN2) reduced intestinal ACE2 mRNA and protein levels in vitro and in vivo. Consistently, we revealed two GCN2 inhibitors, GCN2iB and GCN2-IN-1, downregulated ACE2 protein expression in CCD841 cells. Moreover, we found that increased ACE2 expression in response to leucine deprivation was GCN2 dependent. Through RNA-sequencing analysis, we identified two transcription factors, MAFB and MAFF, positively regulated ACE2 expression under leucine deprivation in CCD841 cells. These findings demonstrate that amino acid deficiency increases ACE2 expression and thereby likely aggravates intestinal SARS-CoV-2 infection.

[1] Zhongshan Hospital, State Key Laboratory of Medical Neurobiology, Institute for Translational Brain Research, MOE Frontiers Center for Brain Science, Fudan University, Shanghai 200032, China. [2] CAS Key Laboratory of Nutrition, Metabolism and Food Safety, Innovation Center for Intervention of Chronic Disease and Promotion of Health, Shanghai Institute of Nutrition and Health, University of Chinese Academy of Sciences, Chinese Academy of Sciences, Shanghai 200031, China. [3] These authors contributed equally: Xiaoming Hu, Yuguo Niu. ✉email: ffguo@fudan.edu.cn

The coronavirus disease-2019 (COVID-19) pandemic continues to present a public health challenge worldwide. Patients with COVID-19 have been reported to exhibit a variety of gastrointestinal symptoms, including diarrhea, nausea, vomiting, gut microbiota alterations, and colonic mucosal injury, with their stool samples testing positive for severe acute respiratory syndrome coronavirus 2 (SARS-CoV-2) RNA[1–4], suggesting the active infection and replication of this virus within the enterocytes. Angiotensin-converting enzyme 2 (ACE2) has recently been confirmed as a primary host cell receptor for SARS-CoV-2, which facilitates viral entry into the cells by recognition and binding via the viral spike glycoprotein[5–7]. Given that ACE2 is highly expressed in intestinal epithelial cells (IECs), it provides a potential primary entry route for SARS-CoV-2 into these cells, thereby causing gastrointestinal symptoms. Therefore, knowledge about the factors that affect or regulate intestinal ACE2 expression may provide key therapeutic targets for reducing gastrointestinal symptoms of COVID-19.

ACE2, a human homolog of ACE, acts as a negative regulator for the renin-angiotensin system (RAS) and catalyzes the cleavage of Angiotensin II into the vasodilator Angiotensin (1–7)[8]. Aside from its regulation of electrolyte homeostasis and blood pressure[9], ACE2 also regulates cardiac contractility and heart morphogenesis[10], pancreatic beta-cell function and glucose homeostasis[11], and mediates proinflammatory signaling[12]. Interestingly, ACE2 has been suggested to play an important role in amino acid metabolism in the intestines[13–15]. Previous studies have shown that ACE2 expression is regulated by a hypoxic status, interferons, and proinflammatory factors[16–18]. Moreover, Sun et al.[19] reported that high-fat diet increased ACE2 protein levels, but not its mRNA levels in the ileum of mice, suggesting that nutrients play an important role in ACE2 expression. Particularly, based on studies on its collectrin domain on IEC plasma membranes, ACE2 has been proven to facilitate amino acid uptake by binding to various amino acid transporters[20,21], and a change in amino acid levels affects the expression of their transporters[22]. Therefore, we speculated that ACE2 might be responsive to different amino acid levels and its expression may be regulated by such changes. Here, this study was carried out to investigate the effects of different amino acid levels on the intestinal ACE2 expression and the underlying mechanisms involved.

## Results

### ACE2 is upregulated in response to amino acid deficiency.
Many patients with COVID-2019 develop gastrointestinal symptoms along with respiratory system disorders[1,2]. As a primary host cell receptor for SARS-CoV-2, ACE2 is reported to be widely expressed in many tissues, with high levels occurring in IECs especially[23,24]. To determine the relationship between ACE2 expression and amino acid level changes, human colonic epithelial CCD841 cells were incubated in either control culture medium or essential amino acid (EAA)-free medium for 48 h. Interestingly, the ACE2 protein levels in the cells were increased upon the deprivation of all EAAs (Supplementary Fig. 1a). To determine which EAA deprivation can regulate ACE2 expression, we further incubated CCD841 cells in culture media lacking different types of EAAs, individually. Firstly, we tested the effects of different amino acid deprivation duration (3, 12, 24, or 48 h) on ACE2 expression. We found that a long period (48 h) leucine deprivation increased ACE2 expression significantly (Fig. 1a and Supplementary Fig. 1b, c). We also found that the mRNA levels of amino acid transporter solute carrier family 1 member 5 (SLC1A5) and asparagine synthase (ASNS) were markedly increased under 48 h leucine deprivation (Supplementary Fig. 1c). Therefore, 48 h of amino acid deprivation was used in all further experiments. Besides isoleucine and histidine, deprivation of other EAAs significantly increased the ACE2 protein levels as well as that of phosphorylated-eukaryotic translation initiation factor 2A (p-EIF2A), a marker of amino acid deficiency[25] (Fig. 1). However, incubation with three-fold amounts of EAAs[26] had no effect on the ACE2 or p-EIF2A expression levels (Supplementary Fig. 2), indicating that ACE2 responded particularly to amino acid deficiency. Interestingly, non-essential amino acid glutamine but not glycine deprivation also induced ACE2 expression in CCD841 cells, which was possibly due to the important role of glutamine in intestinal physiology[27] (Supplementary Fig. 3). Increased ACE2 expression was also detected in colon and subcutaneous white adipose tissue (sWAT) in mice fed leucine deprivation diet for 7 days (Fig. 1b, c and Supplementary Fig. 4). Notably, leucine deprivation in CCD841 cells also induced the expression of another candidate SARS-CoV-2 receptor, tyrosine-protein kinase receptor UFO (AXL)[28] (Supplementary Fig. 5).

Furthermore, as the main symptoms caused by SARS-CoV-2 are respiratory system disorders[29], we also incubated human bronchial epithelial BEAS-2B cells in a medium either free of lysine. The mRNA and protein levels of ACE2 were increased upon lysine deprivation (Supplementary Fig. 6), suggesting that EAA deficiency may promote ACE2 expression in both IECs and respiratory epithelial cells.

### GCN2 regulates ACE2 expression in vitro and in vivo.
GCN2 is the most well-known amino acid sensor, responding to amino acid deficiency at an early stage[30]. To verify its role in the regulation of ACE2 expression under amino acid deficiency conditions, we first analyzed the effects of GCN2 disruption on the expression of the SARS-CoV-2 receptor ACE2 by transfecting CCD841 cells with either small interfering RNAs (siRNAs) targeting the human GCN2 gene (siGCN2) or scramble siRNAs (siNC) for 48 h. The results showed that knockdown of GCN2 significantly decreased ACE2 mRNA levels as well as the p-EIF2A and ACE2 protein levels in the intestinal cells relative to that in control cells (Fig. 2a, b).

To further explore the regulatory role of GCN2 on ACE2 expression in vivo, we constructed IEC-specific Gcn2 knockout (Gcn2 IKO) mice. In these mice, the mRNA and protein levels of GCN2 were negligible in both the small intestine and colon but not changed in lung, stomach, or kidney (Fig. 2c, d and Supplementary Fig. 7a, b), indicating the specificity of villin cre mice. The mRNA and protein levels of ACE2 in the colon were also significantly decreased (Fig. 2c, d). However, although the mRNA of Ace2 decreased in the small intestine of these mice, the protein levels remain unchanged (Fig. 2c, d), suggesting that GCN2 might regulate ACE2 expression in the colon specifically. Collectively, these results indicate that GCN2 regulates ACE2 expression in IECs, especially in the colon.

Notably, we further evaluated the role of three GCN2 inhibitors namely GCN2iB[31], GCN2-IN-1 (compound A-92)[32], and GZD824[33] on ACE2 protein expression in CCD841 cells. Consistent with the effects of GCN2 knockdown, both GCN2iB and GCN2-IN-1 decreased the protein levels of ACE2, whereas GZD824 had no obvious effect on its expression (Fig. 2e and Supplementary Fig. 8).

### Increased ACE2 expression in response to leucine deprivation is dependent on GCN2.
To verify whether the induction of ACE2 expression by leucine deprivation was dependent on GNC2, we first transfected CCD841 cells with either siNC or siGCN2 for 12 h, and then incubated them with either control or leucine-free culture medium for 48 h. Both the mRNA and protein levels of ACE2 were induced by leucine deprivation, whereas these effects

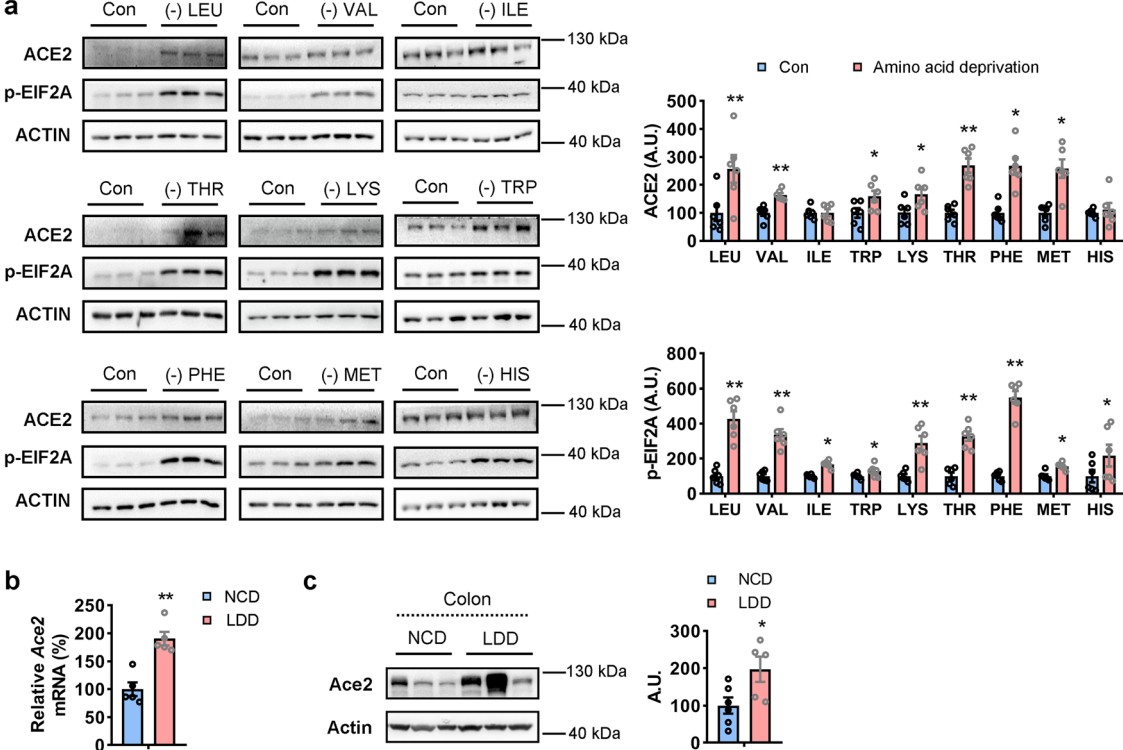

**Fig. 1 ACE2 expression is upregulated in intestinal cells response to amino acid deficiency. a** CCD841 cells were incubated with control culture medium (Con) or essential amino acid (leucine, LEU; Valine, VAL; Isoleucine, ILE; Threonine, THR; Lysine, LYS; Tryptophan, TRP; Phenylalanine, PHE; Methionine, MET; Histidine, HIS) deprived culture medium for 48 h. ACE2 and p-EIF2A expression were analyzed by western blotting (left) and quantified by densitometric analysis (right). **b**, **c** 10-week-old male wild type mice were fed with control diet (NCD) or leucine deprivation diet (LDD) for 7 days. ACE2 mRNA and protein levels in colonic epithelial cells were analyzed by qRT-PCR and western blotting, respectively. Protein levels were quantified by densitometric analysis (right); A.U.: arbitrary unit. Data are expressed as the mean ± SEM ($n$ = 5–6 per group, as indicated by scatter circles). *$P < 0.05$, **$P < 0.01$.

were compromised by *GCN2* knockdown (Fig. 3), indicating that the upregulation of ACE2 expression in response to leucine deprivation was dependent on GCN2.

**GCN2 activates transcription factors for regulating ACE2 expression in response to leucine deprivation**. Having shown that the *ACE2* mRNA levels were increased under leucine deprivation and decreased after *GCN2* knockdown, we speculated that GCN2 regulates several transcription factors that control *ACE2* transcription. Because activating transcription factor 4 (ATF4) is a typical downstream of GCN2[25], we firstly transfected CCD841 cells with either control plasmid or plasmid expressing human *ATF4* for 48 h. We found that overexpression of *ATF4* increased ACE2 expression in CCD841 cells (Supplementary Fig. 9a). Moreover, CCD841 cells were transfected with *siNC* or *siATF4* for 12 h, and then incubated them with either control or glutamine-free culture medium for 48 h. We found that glutamine deprivation-induced ACE2 expression were compromised by *siATF4*, suggesting that ATF4 regulated ACE2 expression under amino acid deprivation (Supplementary Fig. 9b). To explore whether other transcription factors regulated ACE2 expression, we performed RNA-sequencing analysis. Accordingly, we focused on seven candidate transcription factor-encoding genes, whose expression levels were increased upon leucine deprivation (fold change > 1.5, $p$ value < 0.05) but decreased after *GCN2* knockdown (fold change < 0.7, $p$ value < 0.05) (Fig. 4a and Supplementary Fig. 10). To confirm their roles in regulating ACE2 expression further, CCD841 cells were separately transfected with siRNAs targeting each transcription factor. We found that the knockdown of JunB proto-oncogene, AP-1 transcription factor subunit (*JUNB*), RELB proto-oncogene, NF-kB subunit (*RELB*), MAF bZIP transcription factor B (*MAFB*), or

MAF bZIP transcription factor F (*MAFF*) could decrease the mRNA levels of *ACE2* (Fig. 4b, c and Supplementary Fig. 11a–e). Furthermore, when these four transcription factors were individually overexpressed in CCD841 cells, only *MAFB* and *MAFF* could increase the *ACE2* expression levels (Fig. 4d, e and Supplementary Fig. 11f, g). To determine whether *MAFB* and *MAFF* were upstream regulatory factors of ACE2 expression under leucine deprivation, CCD841 cells were transfected with *siNC*, *siMAFB*, or *siMAFF* for 12 h followed by incubation with either control or leucine-free culture medium for 48 h. The knockdown of *MAFB* or *MAFF* blocked the upregulation of ACE2 expression upon leucine deprivation (Fig. 4f, g), suggesting that both transcription factors acted downstream of GCN2. Taken together, these results indicate that the amino acid sensor GCN2 promotes the expression of SARS-CoV-2 receptor ACE2 by activating the expression of key transcription factors, such as ATF4, MAFB, and MAFF, in response to amino acid deficiency.

**Discussion**

The COVID-19 pandemic caused by SARS-CoV-2 continues to pose a major healthcare crisis worldwide, leading to urgency in the need to increase our understanding of the disease pathogenesis, especially the host factors facilitating viral infection. As a primary host cell receptor for SARS-CoV-2, ACE2 variants of gene polymorphisms can modulate susceptibility to this virus[34], and notably, ACE2 is highly expressed on the epithelial cells lining the intestines[23,24], rendering this organ system a potential viral target by facilitating its entry into the IECs, thereby resulting in a series of intestinal symptoms and inflammation[3]. Therefore, we focused on elucidating the underlying mechanisms of ACE2 expression regulation in enterocytes.

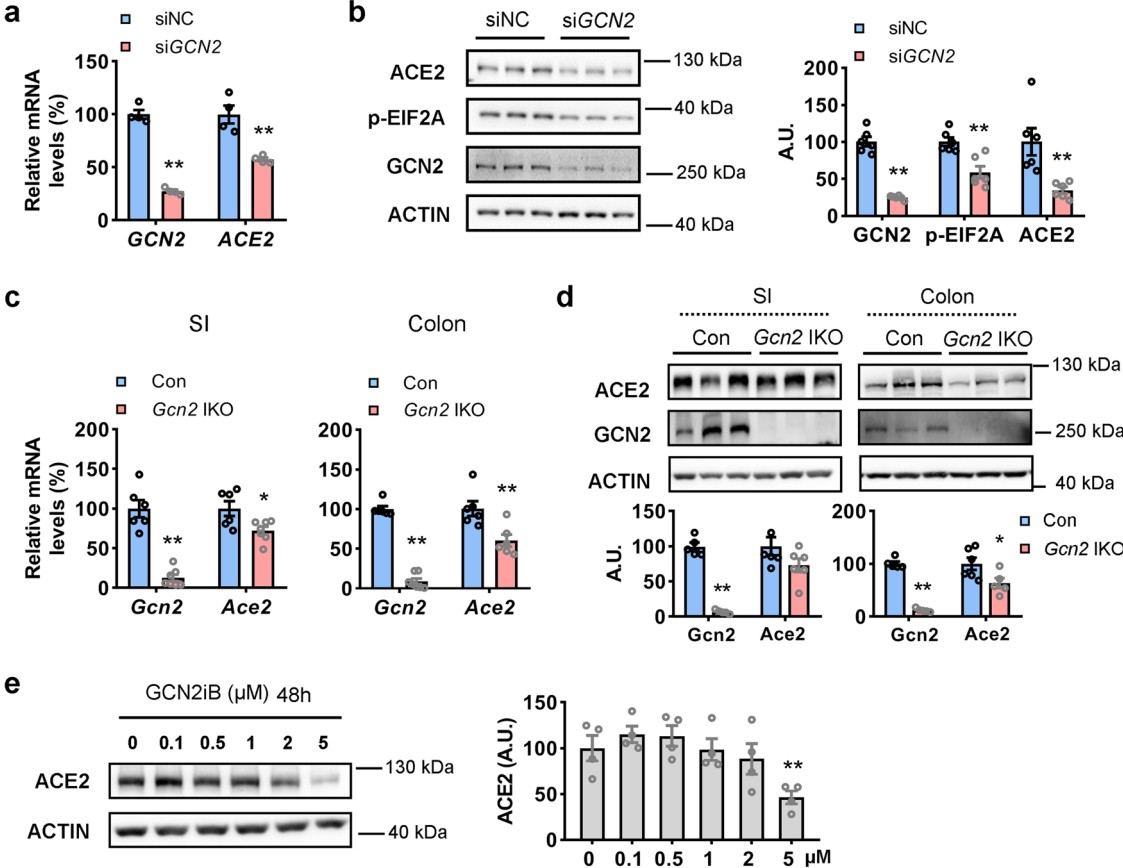

**Fig. 2 GCN2 regulates ACE2 expression in vitro and in vivo. a, b** CCD841 cells were transfected with control small interfering RNAs (siNC) or small interfering RNAs targeting at human *GCN2* (*siGCN2*) for 48 h. mRNA expression of *GCN2* and *ACE2* were analyzed by qRT-PCR (**a**); GCN2, p-EIF2A, and ACE2 protein levels were analyzed by western blotting (left) and quantified by densitometric analysis (right); A.U.: arbitrary unit (**b**); **c, d** small intestinal and colonic epithelial cells isolated from 10-week-old male *Gcn2*-floxed mice (Con) or intestinal epithelial cell (IEC)-specific *Gcn2* deletion mice (*Gcn2* IKO). mRNA expression of *Gcn2* and *Ace2* were analyzed by qRT-PCR (**c**); GCN2 and ACE2 protein levels were analyzed by western blotting (top) and quantified by densitometric analysis (bottom) (**d**). **e** CCD841 cells were incubated with indicated concentration GCN2iB (HY-112654) for 48 h, then ACE2 protein levels were analyzed by western blotting (left) and quantified by densitometric analysis (right). Data are expressed as the mean ± SEM ($n = 3$–6 per group, as indicated by scatter circles). *$P < 0.05$, **$P < 0.01$.

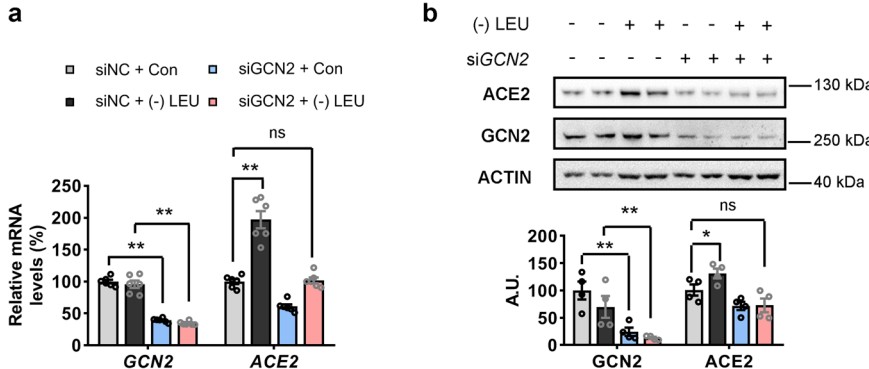

**Fig. 3 Increased ACE2 expression in response to leucine deprivation is dependent on GCN2. a, b** CCD841 cells were transfected with control small interfering RNAs (*siNC*) or small interfering RNAs targeting at human *GCN2* (*siGCN2*). After 12 h, cells were incubated with either control (Con) or leucine-deprived culture medium ((−) LEU) for 48 h. The mRNA and protein expression of *GCN2* and *ACE2* were analyzed by qRT-PCR and western blotting, respectively. Protein quantified by densitometric analysis (right); A.U.: arbitrary unit. Data are expressed as the mean ± SEM ($n = 4$–6 per group, as indicated by scatter circles). *$P < 0.05$, **$P < 0.01$.

Amino acids are one of the three most important nutrients for the human body (the other two being carbohydrates and fats). Importantly, ACE2 has a completely different function independent of the RAS in the gastrointestinal tract[35], where it is responsible for the trafficking of the neutral amino acid transporter B0AT1 to the plasma membrane of IECs[13,36]. Thus, ACE2 regulates the uptake of several amino acids, and its down-regulation leads to significant reductions in tryptophan levels in

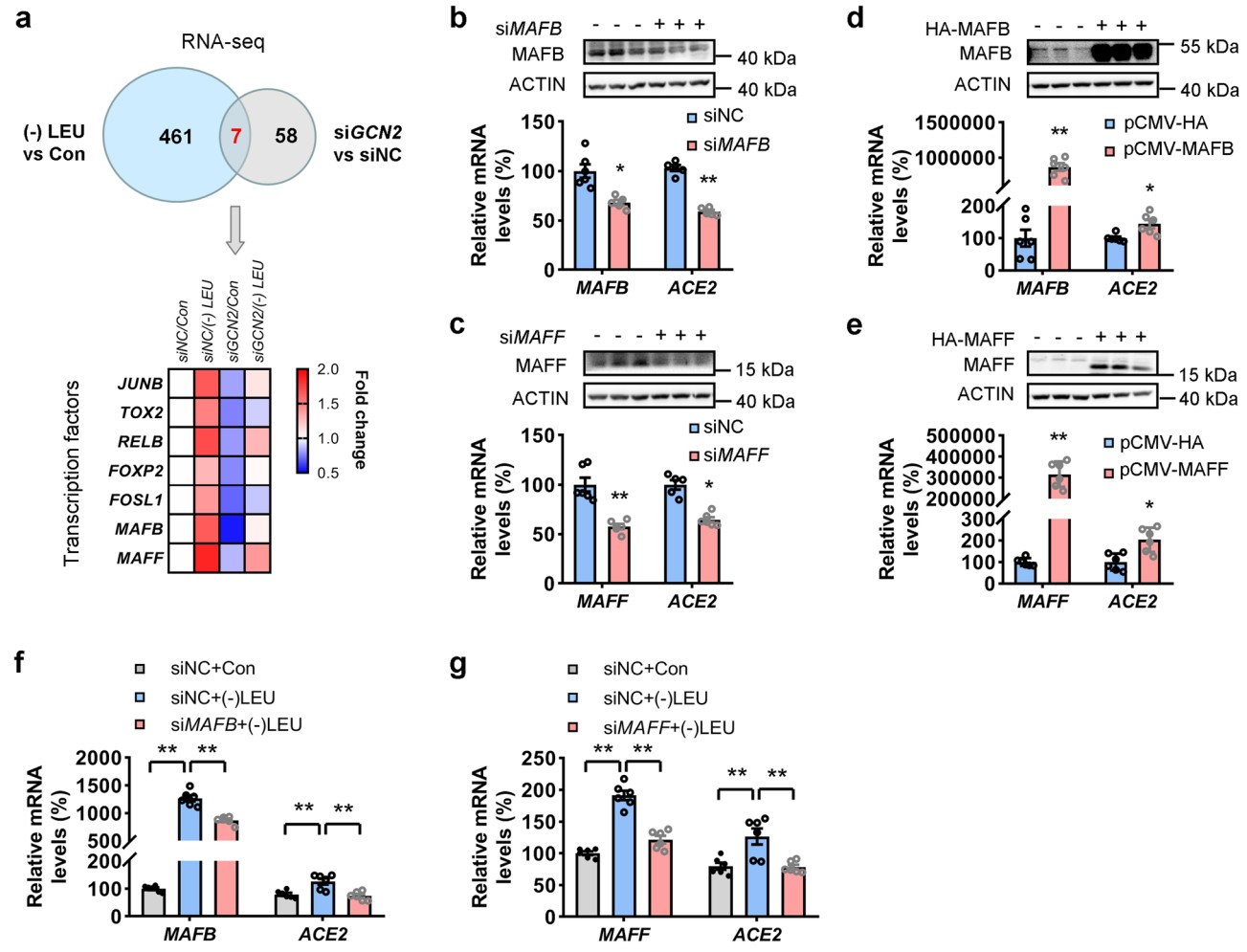

**Fig. 4 Identification of transcription factors regulating ACE2 expression in response to leucine deprivation. a** CCD841 cells were transfected with control small interfering RNAs (*siNC*) or small interfering RNAs targeting at human *GCN2* (*siGCN2*). After 12 h, cells were incubated with either control (Con) or leucine-deprived culture medium ((−) LEU) for 48 h. RNA-seq analysis of differentially expressed transcription factors that were increased under leucine starvation ((−) LEU) (fold change > 1.5, *p* value < 0.05) and decreased by *GCN2* knockdown (fold change < 0.7, *p* value < 0.05). **b–e** CCD841 cells were transfected with *siNC*, *siMAFB*, *siMAFF*, control plasmids (pCMV-HA), HA-tagged MAFB (HA-MAFB), or HA-tagged MAFF (HA-MAFF) for 48 h. MAFB and MAFF protein expression were measured by western blotting (upper); *MAFB*, *MAFF*, and *ACE2* mRNA expression was measured by qRT-PCR (bottom). **f, g** CCD841 cells were transfected with *siNC*, *siMAFB* or *siMAFF*. After 12 h, cells were incubated with either Con or (−) LEU culture medium for 48 h. mRNA expression of *MAFB*, *MAFF*, and *ACE2* were analyzed by qRT-PCR. Data are expressed as the mean ± SEM (*n* = 5–6 per group, as indicated by scatter circles). *P < 0.05, **P < 0.01.

the serum leading to inflammatory bowel disease[14,36]. Previous studies had indicated that amino acid level change could regulate the expression of amino acid transporters[37]; we speculated that such changes could regulate ACE2 expression as well. Interestingly, we found that ACE2 expression was increased in CCD841 cells upon their deprivation of all or single EAAs. We also found that the mRNAs of neutral amino acid transporter *SLC1A5* and *ASNS* were significantly increased under leucine deprivation conditions. We speculated that the upregulation of *SLC1A5* and *ASNS* was compensatory effect, which may aim to keep the balance of amino acid levels. Moreover, we found that both the mRNA and protein levels of ACE2 were increased in human bronchial epithelial BEAS-2B cells upon lysine deprivation. Consistent with in vitro results, ACE2 expression was also increased in colon of mice by leucine deprivation treatment. Moreover, although there was no change in liver ACE2 levels between control and leucine deprivation groups, the ACE2 levels of sWAT were significantly increased in leucine deprivation conditions. Interestingly, previous studies indicated that leucine deprivation can induce sWAT browning[25], and increased ACE2

can induce sWAT browning as well[38]. We speculated that increased ACE2 levels under leucine deprivation treatment might induce this phenomenon. These results suggested that EAA deficiency may promote the expression of this SARS-CoV-2 receptor in the gastrointestinal system, respiratory systems, and specific metabolic organs, such as adipose tissues. In addition to COVID-19, a recent human study reported that ACE2 levels were elevated in colonic ulcerative colitis compared with non-inflammatory bowel disease controls[12]. Therefore, we speculated that intervening amino acid levels or inhibiting GCN2 in colon might also provide a strategy for treating ulcerative colitis.

Previous studies found that acute amino acid deprivation suppressed intestinal inflammation via a mechanism dependent on GCN2, the most important amino acid deficiency sensor[39]. Our current study confirmed that leucine deprivation upregulated ACE2 expression through GCN2 signaling. As ACE2 is a primary host cell receptor for SARS-CoV-2, the downregulation of its expression is desirable in order to inhibit the binding and invasive abilities of the virus effectively. Therefore, GCN2 can be a potential therapeutic target for the ACE2 expression regulation,

providing an antiviral strategy against COVID-19. Furthermore, we revealed that two GCN2 pharmacologic inhibitors, GCN2iB and GCN2-IN-1, could significantly reduce ACE2 protein expression in CCD841 cells, making them candidate drugs for ameliorating the gastrointestinal symptoms in patients with COVID-19. However, it should be noted that the results of our study have been obtained in cell and animal models; thus, further studies are needed to verify the findings in humans.

Recently, Chen et al.[40] identified that four key transcriptional regulators—caudal-type homeobox 2, hepatocyte nuclear factor 4, smad family member 4, and GATA-binding proteins—that could bind to various loci of *ACE2* and regulate its gene expression. Moreover, miRNA-421 and miR-200c were identified as regulators of ACE2 expression inhibition[41,42]. In this study, we revealed two key transcription factors (viz., MAFB and MAFF) that promoted ACE2 expression in the enterocytes. MAFB and MAFF belong to the large- and small-MAF protein members of the family of basic region leucine zipper-type transcription factors, respectively, which are crucial regulators of mammalian gene expression[43,44]. It was reported that MAFB could regulate the expression of a cluster of chemokine-encoding genes that were associated with the COVID-19 cytokine storm[45], suggesting a link between MAFB-regulated genes and the disease pathology. Utilizing the disease-associated database DisGeNet, Vega et al.[46] analyzed the genes that were differentially regulated by MAFB and discovered their significant enrichment in several processes and pathways related to respiratory deterioration, which is a firmly established symptom of COVID-19. These studies suggested that high MAFB expression in lung macrophages might be a critical determinant for the severity and progression of COVID-19. Notably, our study revealed MAFB to be an upstream regulatory factor of ACE2 expression, suggesting that the upregulated expression of MAFB in the intestines may continuously enhance SARS-CoV-2 infection and the ensuing proinflammatory reactions.

In conclusion, the findings of this study suggest that optimal intake of amino acids or foods rich in EAAs may be beneficial for patients with COVID-19, especially in ameliorating the gastrointestinal symptoms related to the disease. The confirmed involvement of the GCN2 signaling pathway in COVID-19 provides potential avenues for altering ACE2 expression in host cells to reduce their susceptibility to SARS-CoV-2 infection or towards the severity of the disease.

## Methods

**Plasmids and siRNAs**. *JUNB*, *RELB*, *MAFB*, and *MAFF* cDNAs were generated from a human cDNA library and cloned into the pCMV-HA vector (Clontech, USA) at the *EcoRI* and *KpnI* sites, respectively. The primers used for polymerase chain reaction (PCR) amplification are listed in Supplementary Table 1. Synthetic siRNA oligonucleotides specific for regions in the human *GCN2*[47], *ATF4*[48], *JUNB*[49], *RELB*[50], *MAFB*[51], *MAFF*[52], *TOX high mobility group box family member 2*[53], *forkhead box P2*[54], and *FOS like 1, AP-1 transcription factor subunit*[55] mRNAs were synthesized by GenePharma (Shanghai, China). The sequences for successful gene knockdown are listed in Supplementary Table 2. CCD841 cells at 50–70% confluency were transfected with the various plasmids or siRNA duplexes (both at a concentration of 1 µg per well in 12-well dishes) using Lipofectamine 3000 (Invitrogen, USA), according to the manufacturer's instructions.

**Cell culture and treatments**. Human colon epithelial CCD841 and bronchial epithelial BEAS-2B cells were originally obtained from the American Type Culture Collection (ATCC, USA). All cells were cultured and maintained in Dulbecco's modified Eagle's medium (high glucose, 4.5 g/L) supplemented with 10% fetal bovine serum, 50 µg/mL streptomycin, and 50 U/mL penicillin according to the ATCC instructions. Media containing complete amino acids (control) or deficient in all EAAs (leucine, valine, isoleucine, lysine, tryptophan, threonine, phenylalanine, methionine, and histidine) or a single EAA were prepared in regular fetal bovine serum-free Dulbecco's modified Eagle's medium. The culture medium formulations were available in Supplementary Table 3.

According to the description in previous studies, appropriate dose of GCN2iB[31], GCN2-IN-1 (A-92)[32], and GZD824[33] (MedChemExpress, USA)

were dissolved in DMSO as stock solutions (final concentration was less than 0.5% solvent).

**Mice, tissue preparation, and IEC isolation**. All mice used in this study were of a C57BL/6J background and maintained under specific pathogen-free conditions. Male C57BL/6J wild-type (WT) mice were purchased from the Shanghai Laboratory Animal Co. Ltd (SLAC; Shanghai, China). *Gcn2*-floxed mice[56] were intercrossed with *Villin*-cre mice[48] (Shanghai Biomodel Organism Science & Technology Development Co.,Ltd., Shanghai, China) to generate the IEC-specific *Gcn2* IKO mice. All mice were housed in laboratory cages at a temperature of 23 ± 3 °C and humidity of 35 ± 5%, under a 12/12 h dark/light cycle, with free access to a regular chow diet (SLAC) and water. Control (nutritionally complete amino acids) and leucine-deficient diets were obtained from Research Diets (New Brunswick, USA). The food formulas of the amino acid diets are listed in Supplementary Table 4. All protocols of animal experiments were performed in accordance with relevant guidelines and regulations and approved by the Institutional Animal Care and Use Committee of Fudan University (Shanghai, China).

The mouse tissue samples were freshly harvested, immediately frozen in liquid nitrogen, and stored at −80 °C for further action. The small intestinal and colonic epithelial cells were isolated according to the method described in a previous study[48]. In brief, the jejunum, ileum, and colon were removed from the sacrificed mice and cut into 0.5 cm thin pieces, and these were then rinsed in cold phosphate-buffered saline to remove debris. Primary IECs were then isolated by incubating the pieces in phosphate-buffered saline containing 2 mmol/L dithiothreitol and 1 mmol/L ethylenediaminetetraacetic acid at 37 °C for 30 min with gentle shaking.

**RNA isolation and quantitative real-time polymerase chain reaction**. Total RNA content was extracted from the tissue and cell samples using the TRIzol reagent (Invitrogen, USA) and treated with RNase-free DNase (MBI Fermentas, Germany). The extracted RNA was then reverse transcribed to cDNA using the RevertAid RT Kit (Invitrogen, USA) according to the manufacturer's instructions. The quantitative real-time PCR was carried out using SYBR Green Master Mix (Invitrogen, USA) on QuantStudio 6 (Applied Biosystems, Foster City, CA, USA). The reaction was carried out according to the following conditions: predenaturation 5 min at 95 °C, 95 °C 10 s followed by 60 °C 30 s for 40 cycles, melting curve stage 95 °C 15 s, 60 °C 60 s, and 95 °C 15 s. The sequences of primers used in this study are listed in Supplementary Table 5. All PCR amplifications were performed in triplicate for each RNA sample, and the expression level of each gene was quantified relative to that of glyceraldehyde 3-phosphate dehydrogenase (GAPDH). The results were analyzed using the $2^{-\Delta\Delta Ct}$ method.

**RNA sequencing and data analysis**. Total RNA content was extracted from CCD841 cells using the TRIzol reagent (Invitrogen, USA) according to the manufacturer's instructions, and the integrity and quality were checked using a NanoDrop 2000 spectrophotometer and formaldehyde agarose gel electrophoresis. Only RNA samples with a ratio of absorbance at 260 nm to that at 280 nm ($A_{260\ nm}/A_{280\ nm}$) greater than 1.8 were used. For RNA sequencing library synthesis, 3 µg of total RNA was first depleted of rRNA using the Ribo-Zero rRNA Magnetic Kit (Plant Seed/Root kit, Epicenter, Madison, USA). Sequencing libraries were generated using NEBNext® UltraTM RNA Library Prep Kit for Illumina® (NEB, USA) following manufacturer's recommendations. Sequencing was then performed on an Illumina HiSeq 2500 platform and 150 bp paired-end reads were generated.

Raw data (raw reads) of fastq format were first processed through in-house perl scripts. In this step, clean data (clean reads) were obtained by removing reads containing adapter, reads containing ploy-N and low-quality reads from raw data. The clean reads were aligned to the Ensembl 70 gene annotation of the NCBI38/mm10 genome using Bowtie with default parameters. FeatureCounts v1.5.0-p3 was used to count the reads numbers mapped to each gene. And then FPKM of each gene was calculated based on the length of the gene and reads count mapped to this gene. Differential expression analysis of two conditions/groups was performed using the DESeq2 R package[57]. DESeq2 provides statistical routines for determining differential expression in digital gene expression data using a model based on the negative binomial distribution. The resulting *P*-values were adjusted using the Benjamini and Hochberg's approach for controlling the false discovery rate. Genes with an adjusted *P*-value < 0.05 found by DESeq2 were assigned as differentially expressed. The accession number for the sequencing data reported in this paper is SRA: PRJNA808827.

**Western blot analysis**. Protein samples were extracted from the samples using a protein lysis buffer [10 mM 4-(2-hydroxyethyl)-1-piperazineethanesulfonic acid (pH 7.6), 1.5 mM MgCl₂, 0.5 mM dithiothreitol, 10 mM KCl, 10 mM NaF, 1 mM Na3VO4, and 0.5 mM phenylmethylsulfonyl fluoride] supplemented with a protease inhibitor cocktail according to the manufacturer's protocol. After determination of the protein concentration, the samples were subjected to western blot analysis. Antibodies used for western blotting were listed in Supplementary Table 6. The specific proteins were visualized by enhanced chemiluminescence using ECL Plus (Amersham Biosciences, UK). The band intensities were quantitated using Quantity One (Bio-Rad Laboratories) and normalized to that of

beta-actin. The number of replicates analyzed in western blot was as indicated by scatter circles in statistical histogram.

**Statistics and reproducibility**. Statistical analysis of the data was performed using GraphPad Prism version 8.0 (San Diego, CA, USA). All values are presented as the means ± standard error of the means ($n$ as indicated in the figure legends). Statistical comparisons of two groups were performed using the unpaired two-tailed Student $t$-test; data involving more than two groups were assessed using an ANOVA followed by Bonferroni's post hoc correction. Significant differences between data are indicated in the figures by *$P < 0.05$, **$P < 0.01$.

**Reporting summary**. Further information on research design is available in the Nature Research Reporting Summary linked to this article.

## Data availability

The source data underlying the graphs and charts in the main manuscript file are shown in Supplementary Data 1. RNA-sequencing data from this study have been deposited online and available at https://www.ncbi.nlm.nih.gov/bioproject/PRJNA808827/. Correspondence and requests for materials should be addressed to Feifan Guo.

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

## Acknowledgements

This work was supported by grants from the National Natural Science Foundation (91957207, 31830044, 81870592, 82170868, 81970731, 81970742, 81770852, and 82000764), the National Key R&D Program of China (2018YFA0800600), Shanghai leading talent program, CAS Interdisciplinary Innovation Team, Novo Nordisk-Chinese Academy of Sciences Research Fund (NNCAS-2008-10), and the Natural Science Foundation of Shanghai "science and technology innovation action plan"(21ZR1475900). Xiaoming Hu was supported by Youth Innovation Promotion Association CAS.

## Author contributions

X.H., Y.N., P.L., F.X., F.Y., and H.Y. performed the experiments. X.H. and Y.N. designed experiments, developed analysis tools, analyzed data, and wrote the paper. S.C. and F.G. interpreted the results. F.G. contributed to the design of the experiments, conceived of the study, and wrote the paper. All authors have read and commented on the manuscripts.

## Competing interests

The authors declare no competing interests.
