## [Peer Review File · Communications Biology]

Reviewer #1 (Remarks to the Author):

The current manuscript investigated the expression of ACE2, identified as a primary receptor for severe acute respiratory syndrome coronaviruses 2 (SARS-CoV-2) in enterocytes. By treating CCD841 cells with deprivation of all or one essential amino acid, they showed that ACE2 expression was upregulated. Using mice with GCN2 knockdown or inhibitor for GCN2, they found that activation of GCN2 played an important role in ACE2 expression. RNA-seq analysis identified that two novel transcription factors, MAFB and MAFF, stimulated ACE2 expression under leucine deprivation in CCD841 cells. Together, these data demonstrated that ACE2 expression was increased under amino acid deficiency that may aggravate SARS-CoV-2 infection in intestine. These results shed new light on the understanding the upstream signal regulating ACE2, and the entry of the virus during infection. A few comments.

1. As an important downstream of GCN2 following deprivation of amino acid, the role of ATF4 in the regulation of ACE2 expression should not be ignored. Do the authors have evidence that ATF4 regulates ACE2 expression under amino acid deprivation?
2. It's known that ACE2 regulates amino acid transporter. Amino acid deprivation will change expression of transporter, it will be interesting to evaluate the changes of related amino acid transporter in the study, or the relevant information should be discussed.
3. Figure 4A, the authors should show the fold change of the differentially expressed transcriptional factors.
4. Does the regulatory effect of amino acid deprivation on ACE2 is specific just for intestine or can be generalized to other tissues as well? The authors need to discuss this important issue.

Reviewer #2 (Remarks to the Author):

In this study, the authors investigated the regulation of ACE2 expression in intestinal epithelial cells under amino acid deprivation conditions. They found that ACE2 expression was significantly influenced by the protein building block amino acid and that its expression was tightly regulated in a GCN2-dependent manner. Importantly, two novel transcription factors, MAFB and MAFF, were found to be associated with the regulation of ACE2 expression under deprivation of an essential amino acid. These results provide novel and important perspective for better understanding nutritional control of ACE2 expression. This study suggests that optimal intake of amino acids in diets may be beneficial for preventing COVID-19 infection. These data are very interesting. However, several points require to be addressed as below:

1. Whether was ACE2 expression regulated by essential amino acids or leucine deprivation in vivo?
2. How about the effect of non-essential amino acid on ACE2 expression?
3. Since the regulation of ACE2 is dependent on GCN2, how about the effect of GCN2 inhibitor on the expression of MAFB?
4. How about the protein levels of MAFB and MAFF?
5. The efficiency of siRNA used in the study should be validated or reference should be provided to demonstrate the efficiency.
6. Please check the accuracy of the Fig 1A. It appears that the expression of b-actin was missing.

Reviewer #3 (Remarks to the Author):

In this manuscript, Hu et. al. report their findings on ACE2 regulation in enterocytes. The study identifies amino acid deprivation as a factor upregulating ACE2 protein expression and its dependence

on GCN2. The authors base their conclusions on human cell line and primary murine cell culture data and acknowledge that limitation, however, the study includes new information and can serve as a base for future projects. Several changes are strongly encouraged to enhance the manuscript's flow and reproducibility.

Major points

1. The manuscript would benefit from more careful wording when assessing significance of reported findings. For example:

a) (lines 25-27) 'Consistently, we revealed two GCN2 inhibitors, GCN2iB and GCN2-IN-1, downregulated ACE2 protein expression in CCD841 cells in a dose-dependent manner.' -> Only the highest dose of GCN2iB reaches statistical significance in reducing ACE2 level as per figure 2E. Three doses out of five seem to not change ACE2 levels. Similarly, for GCN2-IN-1 as per figure S5A, ACE2 level remains stable or increases at low doses, then is equally decreased for the two highest concentrations. As a minor point, there is a discrepancy in results as the same timing and dosage of GCN2-IN-1 in figures S5A and B (12h, 50µM) results in about 20 and 40 percent decrease in ACE2 expression (this can be in part explained by the fact that they are compared to different control samples).

b) (lines 117-121) 'mRNA and protein levels of Ace2 in the colon were also significantly decreased (Fig. 2C and D). However, although the mRNA of Ace2 decreased in the small intestine of these mice, the protein levels remained unchanged (Fig. 2C and D), suggesting that GCN2 might regulate ACE2 expression in the colon specifically.' -> wide spread of datapoints for colon ACE2 in control mice (figure 2D) should be addressed, as it may be the cause for statistical significance in the panel

2. Reproducibility of the manuscript would be greatly enhanced if the authors were able to provide additional methodological details, such as:

a) catalogue numbers of antibodies used for Western blot, as well as a full gel photographs for validation of appropriate molecular sizes – this is especially important for ACE2, as recent literature (Onabajo et al., Nature Genetics 2020, PMID 33077916) revealed presence of ACE2 isoforms regulated independently of the full-length protein

b) concentrations of used siRNAs and plasmids

c) concentrations of amino acids used to obtain results in figure S2

3. RNA-seq data deposited in SRA could not be found at the time of the review using accession number provided in the manuscript, thus its quality could not be assessed.

4. Figure 1A does not add novelty to the main findings of the manuscript. There already is evidence for high ACE2 expression in the intestine (Hikmet et al., Molecular Systems Biology 2020, PMID 32715618; Li et. al, Infectious Diseases of Poverty 2020, PMID 32345362), and due to the high exposure intensity and widely varied actin detection, normalization of ACE2 to beta-actin signal seems unreliable if the blot presented is representative.

Minor points

1. The claim of obtaining IEC-specific GCN2 KO mouse (lines 115, 267) could be additionally supported by showing lack of off-target effects of the villin cre (for example by positive GCN2 ICH staining or qPCR in renal tissue). While levels of renal GCN2 may not impact this study, if the gut specificity is real, such mice would be a valuable resource for research community.

2. Additional references would bolster the manuscript, both to support claims like (line 90) '2A (p-

EIF2A), a marker of amino acid deficiency', which may not be obvious for people outside the field, and to add details regarding methods, like the rationale for using threefold concentrations of amino acids or specific doses and timepoints for GCN2 inhibitor studies.

3. The manuscript relates its findings to understanding and treating COVID-19. Discussing the results in relation to potential human studies outside the current pandemic would be beneficial for rising the manuscript's significance.

4. Description of figures 4 and S7 in the main text does not entirely match the panels of the figures.

5. AXL (figure S3) is not referenced in the main text as such; adding this gene name in line 95 could be helpful for the reader.

6. Figure S5A has a different Y-scale than all the other bar plots, with 100% mark not labeled.

7. Abbreviations section is missing GCN2.

Point-by-point response to the referees' comments:

Reviewer #1: The current manuscript investigated the expression of ACE2, identified as a primary receptor for severe acute respiratory syndrome coronaviruses 2 (SARS-CoV-2) in enterocytes. By treating CCD841 cells with deprivation of all or one essential amino acid, they showed that ACE2 expression was upregulated. Using mice with GCN2 knockdown or inhibitor for GCN2, they found that activation of GCN2 played an important role in ACE2 expression. RNA-seq analysis identified that two novel transcription factors, MAFB and MAFF, stimulated ACE2 expression under leucine deprivation in CCD841 cells. Together, these data demonstrated that ACE2 expression was increased under amino acid deficiency that may aggravate SARS-CoV-2 infection in intestine. These results shed new light on the understanding the upstream signal regulating ACE2, and the entry of the virus during infection.

A few comments.

1. As an important downstream of GCN2 following deprivation of amino acid, the role of ATF4 in the regulation of ACE2 expression should not be ignored. Do the authors have evidence that ATF4 regulates ACE2 expression under amino acid deprivation?

Our response:

We agree with the reviewer that ATF4 is a typical downstream of GCN2 (Yuan *et al.*, *Nat. Commun.* 2020). To answer this question, we performed the following experiments: CCD841 cells were transfected with control plasmid or plasmid expressing human *ATF4* for 48 h. We found that overexpression of ATF4 increased ACE2 expression in CCD841 cells (Figure S1A). Moreover, CCD841 cells were transfected with control small interfering RNAs or small interfering RNAs targeting at human *ATF4* (*siATF4*). After 12 h, cells were incubated with either control or glutamine deprivation culture medium for 48 h. We found that glutamine deprivation induced ACE2 expression was reversed by *siATF4* (Figure S1B). These results suggested that ATF4 regulated ACE2 expression under amino acid deprivation.

This information has been added to Results (page 8) and Supplementary information (Supplementary Fig. 9) in the revised manuscripts.

Figure S1. ATF4 regulates ACE2 expression under amino acid deprivation *in vitro*.

(A) and (B) ATF4 and ACE2 protein levels were analyzed by western blotting (left) and quantified by densitometric analysis (right); A.U.: arbitrary unit. CCD841 cells were transfected with control plasmid (Con) or plasmid expressing human *ATF4* (*ATF4* OE) for 48 h in (A). CCD841 cells were transfected with control small interfering RNAs (*siNC*) or small interfering RNAs targeting at human *ATF4* (*siATF4*). After 12 h, cells were incubated with either control (+ GLN) or glutamine deprivation (- GLN) culture medium for 48 h in (B). Data are expressed as the mean \pm SEM (n = 3–6 per group, as indicated). * $P < 0.05$; ** $P < 0.01$.

2. It's known that ACE2 regulates amino acid transporter. Amino acid deprivation will change expression of transporter, it will be interesting to evaluate the changes of related amino acid transporter in the study, or the relevant information should be discussed.

Our response:

We agree with the reviewer's opinions. We detected the expression of neutral amino acid transporter *SLC1A5* and asparagine synthase *ASNS* in CCD841 cells under leucine deprivation conditions, and found that the mRNAs of *ACE2*, *SLC1A5* and *ASNS* were significantly increased (Figure S2). We speculated that the upregulation of *SLC1A5* and *ASNS* was compensatory effect, which may aim to keep the balance of amino acid levels.

We have added this information to Results (page 5), Discussion (page 10), and Supplementary information (Supplementary Fig. 1C) in our revised manuscripts.

Figure S2. The expression of *SLC1A5* and *ASNS* under leucine deprivation conditions.

mRNA expression of *ACE2*, solute carrier family 1 member 5 (*SLC1A5*), and asparagine synthase (*ASNS*) were analyzed by qRT-PCR. Studies were conducted using CCD841 cells, which incubated with either control (Con) or leucine deprivation [(-) LEU] culture medium for 48 h. Data are expressed as the mean \pm SEM (n = 6 per group, as indicated). ** $P < 0.01$.

3. Figure 4A, the authors should show the fold change of the differentially expressed transcriptional factors.

Our response:

Thank you for pointing out this issue. We focused on seven candidate transcription factor-encoding genes, whose expression levels were increased upon leucine deprivation

(fold change > 1.5, p value < 0.05) and decreased after GCN2 knockdown (fold change < 0.7, p value < 0.05).

We have added this information to Results (page 8) and Figure legends (Figure 4A) in our revised manuscripts.

4. Does the regulatory effect of amino acid deprivation on ACE2 is specific just for intestine or can be generalized to other tissues as well? The authors need to discuss this important issue.

Our response:

Thank you for your suggestion. To conform whether the regulatory effect of amino acid deprivation on ACE2 can be generalized to other tissues, we detected the expression of ACE2 in the colon, liver and sWAT in mice fed a 7-day normal chow diet (NCD) or leucine deprivation diet (LDD). Consistent with *in vitro* results, LDD also increased ACE2 expression in colon (Figure S3). Interestingly, although there was no change in liver ACE2 levels between NCD and LDD groups, the ACE2 levels of sWAT were increased in LDD group compared to NCD group (Figure S3). Reportedly, LDD can induce sWAT browning (Yuan *et al.*, *Nat. Commun.* 2020), and increased ACE2 can induce sWAT browning as well (Cao *et al.*, *Elife.* 2022). We speculated that increased ACE2 levels under LDD treatment might induce this phenomenon.

This information has been added to Results (page 5), Discussion (page 10-11), Figure (Fig. 1C) and Supplementary information (Supplementary Fig. 4) in the revised manuscripts.

Figure S3. ACE2 expression in related tissues in response to leucine deprivation *in vivo*.

10-week-old male wild type mice were fed with control diet (NCD) or leucine deprivation diet (LDD) for 7 days. ACE2 protein levels in colon, liver, and sWAT were analyzed by western blotting (left) and quantified by densitometric analysis (right); A.U.: arbitrary unit. Data are expressed as the mean \pm SEM ($n = 5-6$ mice per group, as indicated). * $P < 0.05$, ** $P < 0.01$.

Reviewer #2 (Remarks to the Author):

In this study, the authors investigated the regulation of ACE2 expression in intestinal epithelial cells under amino acid deprivation conditions. They found that ACE2 expression was significantly influenced by the protein building block amino acid and that its expression was tightly regulated in a GCN2-dependent manner. Importantly, two novel transcription factors, MAFB and MAFF, were found to be associated with the regulation of ACE2 expression under deprivation of an essential amino acid. These results provide novel and important perspective for better understanding nutritional control of ACE2 expression. This study suggests that optimal intake of amino acids in diets may be beneficial for preventing COVID-19 infection. These data are very interesting. However, several points require to be addressed as below:

1. Whether was ACE2 expression regulated by essential amino acids or leucine deprivation *in vivo*?

Our response:

This is a good suggestion. To answer this question, we detected the expression of ACE2 in colon, liver and sWAT of mice fed either a 7-day NCD or LDD. Consistent with *in vitro* results, LDD also increased ACE2 expression in colon (Figure S4). Interestingly, although there was no change in liver ACE2 levels between NCD and LDD groups, the ACE2 levels of sWAT were increased in LDD group compared to NCD group (Figure S4). Reportedly, LDD can induce sWAT browning (Yuan *et al.*, *Nat. Commun.* 2020), and increased ACE2 can induce sWAT browning as well (Cao *et al.*, *Elife.* 2022). We speculated that increased ACE2 levels under LDD treatment might induce this phenomenon.

This information has been added to Results (page 5), Discussion (page 10), Figure (Fig. 1C) and Supplementary information (Supplementary Fig. 4) in the revised manuscripts.

Figure S4. ACE2 expression in related tissues in response to leucine deprivation *in vivo*.

10-week-old male wild type mice were fed with control diet (NCD) or leucine deprivation diet (LDD) for 7 days. ACE2 protein levels in colon, liver, and sWAT were analyzed by western blotting (left) and quantified by densitometric analysis (right); A.U.: arbitrary unit. Data are expressed as the mean \pm SEM ($n = 5-6$ mice per group, as indicated). * $P < 0.05$, ** $P < 0.01$.

2. How about the effect of non-essential amino acid on ACE2 expression?

Our response:

Thank you for your suggestion. We chose two non-essential amino acid, glutamine and glycine, to answer this question. CCD841 cells were incubated with control, glutamine deprivation, or glycine deprivation culture medium for 48 h. We found that glutamine, but not glycine, deprivation increased ACE2 expression (Figure S5). It is possibly that glutamine is the main respiratory substrate of IECs and plays an important role in intestinal physiology (Newsholme *et al.*, *J Nutr Biochem.* 1999).

This information has been added to Results (page 5) and Supplemental information (Supplementary Fig. 3) in the revised manuscripts.

Figure S5. Non-essential amino acid deprivation on ACE2 expression *in vitro*.

(A) and (B) ACE2 and p-EIF2A protein levels were analyzed by western blotting (left) and quantified by densitometric analysis (right); A.U.: arbitrary unit. CCD841 cells were incubated with control (Con), glutamine deprivation ((-) GLN), or glycine deprivation ((-) GLY) culture medium for 48 h in (A) and (B). Data are expressed as the mean \pm SEM (n = 5–6 per group, as indicated). ** $P < 0.01$.

3. Since the regulation of ACE2 is dependent on GCN2, how about the effect of GCN2 inhibitor on the expression of MAFB?

Our response:

Thank you for your suggestion. In our work, we found that GCN2 increased ACE2 expression by elevating transcription factors, such as MAFB and MAFF. Therefore, we examined MAFB and MAFF levels in CCD841 cells, which were treated with GCN2 inhibitors at the indicated concentration for the indicated time. Consistently, both GCN2iB and GCN-IN-1 decreased MAFB and MAFF levels (Figure S6).

Figure S6. GCN2 inhibitors decrease MAFB and MAFF expression *in vitro*.

(A – C) MAFB and MAFF protein levels were analyzed by western blotting (left) and quantified by densitometric analysis (right); A.U.: arbitrary unit. CCD841 cells were incubated with indicated concentration of GCN2ib or GCN-IN-1 for the indicated time. Data are expressed as the mean \pm SEM ($n = 3\text{--}6$ per group, as indicated). * $P < 0.05$, ** $P < 0.01$.

4. How about the protein levels of MAFB and MAFF?

Our response:

Thank you for your suggestion. We examined the protein levels of MAFB and MAFF in CCD841 cells, which were transfected with control plasmids, or plasmids expressing HA-tagged MAFB or HA-tagged MAFF. We found that the protein levels of both MAFB and MAFF were increased (Figure S7A and S7B). Moreover, we also detected the protein levels of MAFB or MAFF in CCD841 cells, which were transfected with *siNC*, or small interfering RNAs targeting at human *MAFB* (*siMAFB*) or human *MAFF* (*siMAFF*). The protein levels of both MAFB and MAFF were decreased, accordingly (Figure S7C and S7D).

This information has been added to Results (page 8 and 9) and Figures (4B-4E) in the revised manuscripts.

Figure S7. MAFB and MAFF protein levels.

(A–D) MAFB and MAFF protein levels were analyzed by western blotting (top) and quantified by densitometric analysis (bottom); A.U.: arbitrary unit. CCD841 cells were transfected with control plasmids, plasmids expressing HA-tagged MAFB (HA-MAFB), or HA-tagged MAFF (HA-MAFF) for 48 h in (A) and (B). CCD841 cells were transfected with control small interfering RNAs (*siNC*), small interfering RNAs targeting at human *MAFB* (*siMAFB*), or human *MAFF* (*siMAFF*) for 48 h in (C) and (D). Data are expressed as the mean \pm SEM ($n = 4\text{--}6$ per group, as indicated). * $P < 0.05$, ** $P < 0.01$.

5. The efficiency of siRNS used in the study should be validated or reference should be provided to demonstrate the efficiency.

Our response:

Thank you for pointing out this issue. We have added related references to Methods (page 13) in our revised manuscripts.

6. Please check the accuracy of the Fig 1A. It appears that the expression of b-actin was missing.

Our response:

Thank you for pointing out this issue. In fact, the expression profile of ACE2 has been reported in previous studies (Hikmet *et al.*, *Mol. Syst. Biol.* 2020; Li *et al.*, *Infect Dis Poverty.* 2020). To avoid confusion, we have removed this panel and added related references in our revised manuscripts (Please also see our response to question No. 4, from reviewer 3).

Reviewer #3 (Remarks to the Author):

In this manuscript, Hu et. al. report their findings on ACE2 regulation in enterocytes. The study identifies amino acid deprivation as a factor upregulating ACE2 protein expression and its dependence on GCN2. The authors base their conclusions on human cell line and primary murine cell culture data and acknowledge that limitation, however, the study includes new information and can serve as a base for future projects. Several changes are strongly encouraged to enhance the manuscript's flow and reproducibility.

Major points

1. The manuscript would benefit from more careful wording when assessing significance of reported findings. For example:

a) (lines 25-27) 'Consistently, we revealed two GCN2 inhibitors, GCN2iB and GCN2-IN-1, downregulated ACE2 protein expression in CCD841 cells in a dose-dependent manner.' -> Only the highest dose of GCN2iB reaches statistical significance in reducing ACE2 level as per figure 2E. Three doses out of five seem to not change ACE2 levels. Similarly, for GCN2-IN-1 as per figure S5A, ACE2 level remains stable or increases at low doses, then is equally decreased for the two highest concentrations. As a minor point, there is a discrepancy in results as the same timing and dosage of GCN2-IN-1 in figures S5A and B (12h, 50µM) results in about 20 and 40 percent decrease in ACE2 expression (this can be in part explained by the fact that they are compared to different control samples).

Our response:

We agree with the reviewer that some sentences in the manuscripts need more careful wording. We have replaced these sentences 'Consistently, we revealed two GCN2 inhibitors, GCN2iB and GCN2-IN-1, downregulated ACE2 protein expression in CCD841 cells in a dose-dependent manner' with 'Consistently, we revealed two GCN2 inhibitors, GCN2iB and GCN2-IN-1, downregulated ACE2 protein expression in CCD841 cells'.

Moreover, we repeated the following experiments: CCD841 cells were incubated with 0, 1, 5, 10, 50, or 100 µM GCN2-IN-1 for 12 h, or incubated with 50 µM GCN2-IN-1 for 0, 6, 12, or 24 h. We found that the same timing and dosage of GCN2-IN-1 (50 µM, 12 h) has similar effect on the decrease of ACE2 expression (about 40% decrease) (Figure S8).

This information has been added to Results (page 7) and Supplementary information (Supplementary Fig. 8) in the revised manuscripts.

Figure S8. The effects of GCN2 inhibitors on ACE2 expression *in vitro*.

(A) and (B) ACE2 protein levels were analyzed by western blotting (left) and quantified by densitometric analysis (right); A.U.: arbitrary unit. CCD841 cells were incubated with indicated concentration of GCN2-IN-1 for the indicated time. Data are expressed as the mean \pm SEM (n = 3–6 per group, as indicated). * $P < 0.05$, ** $P < 0.01$.

b) (lines 117-121) 'mRNA and protein levels of Ace2 in the colon were also significantly decreased (Fig. 2C and D). However, although the mRNA of Ace2 decreased in the small intestine of these mice, the protein levels remained unchanged (Fig. 2C and D), suggesting that GCN2 might regulate ACE2 expression in the colon specifically.' - > wide spread of datapoints for colon ACE2 in control mice (figure 2D) should be addressed, as it may be the cause for statistical significance in the panel.

Our response:

Thank you for your suggestion. To answer this question, we repeated this experiment and obtained results that ACE2 was decreased in the colon of IEC-specific *Gcn2* deletion (*Gcn2* IKO) mice (Figure S9), which was consistent with the results in our original manuscripts.

This information has been added to Figures (Fig. 2D) in our revised manuscripts.

Figure S9. ACE2 expression is decreased in intestinal epithelial cells-specific *Gcn2* deletion (*Gcn2* IKO) mice.

ACE2 protein levels were analyzed by western blotting (top) and quantified by densitometric analysis (bottom); A.U.: arbitrary unit. Colonic epithelial cells were isolated

from 10-week-old male *Gcn2*-floxed mice (Con) or *Gcn2* IKO mice. Data are expressed as the mean \pm SEM ($n = 5-6$ mice per group, as indicated). * $P < 0.05$.

2. Reproducibility of the manuscript would be greatly enhanced if the authors were able to provide additional methodological details, such as:

a) catalogue numbers of antibodies used for Western blot, as well as a full gel photographs for validation of appropriate molecular sizes – this is especially important for ACE2, as recent literature (Onabajo *et al.*, Nature Genetics 2020, PMID 33077916) revealed presence of ACE2 isoforms regulated independently of the full-length protein.

Our response:

Thank you for your suggestion. We have added the catalogue numbers of antibodies used for western blot to Supplementary information (Supplementary Table 4) in our revised manuscripts.

In fact, in order to validate the effects of two ACE2 antibodies (from CST and Abclone) and confirm the band of ACE2 appropriate molecular sizes, we transfected negative control siRNAs (siNC) or human *ACE2* siRNAs (si*ACE2*) into CCD841 cells for 48 h. We found that the band of ACE2 protein at approximately 130 kDa was markedly decreased by si*ACE2* (Figure S10A and B), while increased by leucine deprivation (Figure S10C). Moreover, we found that the effect of Abclone's ACE2 antibody was more excellent than CST, therefore, we selected Abclone antibody and ~130 kDa band for further analysis.

Figure S10. Effects of ACE2 antibodies validation.

(A) and (B) CCD841 cells were transfected with negative control siRNA (siNC) or siRNAs targeting at human *ACE2* (si*ACE2*) for 48 h. ACE2 protein levels were analyzed by western blotting (left) using CST and Abclone antibodies, respectively. (C) ACE2 protein levels were analyzed by western blotting (left) using Abclone antibody. CCD841 cells were incubated with control (Con) or leucine deprivation [(-) LEU] culture medium for 48 h. The right panes were ACE2 membrane were exposure in white light.

Moreover, we have added full gel photographs (Original photos) and indicated molecular sizes to all western blotting gels (Figures and Supplementary Figures) in our revised manuscripts.

b) concentrations of used siRNAs and plasmids

Our response:

Thank you for your suggestion. The concentration of siRNA or plasmids used in this study was 1 µg/well in 12-well dishes.

We have added this information to Methods (page 13) in our revised manuscripts.

c) concentrations of amino acids used to obtain results in figure S2

Our response:

Thank you for your suggestion. We have added the culture medium formulations, which included the concentration of amino acid, to Supplementary information (Supplementary Table 5) in our revised manuscripts.

3. RNA-seq data deposited in SRA could not be found at the time of the review using accession number provided in the manuscript, thus its quality could not be assessed.

Our response:

We are sorry for this confuse. We have added the correct accession number to the Method section (page 17 and 19) in our revised manuscripts.

4. Figure 1A does not add novelty to the main findings of the manuscript. There already is evidence for high ACE2 expression in the intestine (Hikmet *et al.*, Molecular Systems Biology 2020, PMID 32715618; Li *et. al.*, Infectious Diseases of Poverty 2020, PMID 32345362), and due to the high exposure intensity and widely varied actin detection, normalization of ACE2 to beta-actin signal seems unreliable if the blot presented is representative.

Our response:

We agree with the reviewer that intestine has been reported to express high levels of ACE2 (Hikmet *et al.*, *Mol. Syst. Biol.* 2020; Li *et al.*, *Infect Dis Poverty.* 2020). To avoid confusion, we have removed related data and replaced with references as the reviewer mentioned in this comment.

This information has been added to Results (page 4) in our revised manuscripts.

Minor points

1. The claim of obtaining IEC-specific GCN2 KO mouse (lines 115, 267) could be additionally supported by showing lack of off-target effects of the villin cre (for example by positive GCN2 ICH staining or qPCR in renal tissue). While levels of renal GCN2 may not impact this study, if the gut specificity is real, such mice would be a valuable resource for research community.

Our response:

Thank you for your suggestion. To answer this question, we detected GCN2 expression in lung, stomach, and renal tissue of *Gcn2*-floxed or *Gcn2* IKO mice. We found that GCN2 levels in these tissues had no difference between the two groups (Figure S11), suggesting that villin cre used in our work had no off-target effects.

This information has been added to Results (page 6) and Supplementary information (Supplementary Fig. 7) in our revised manuscripts.

Figure S11. GCN2 expression in lung, stomach, and kidney of intestinal epithelial cells-specific *Gcn2* deletion (*Gcn2* IKO) mice.

GCN2 protein levels were analyzed by western blotting (left) and quantified by densitometric analysis (right); A.U.: arbitrary unit. Studies were conducted using 10-week-old male *Gcn2*-floxed (Con) or *Gcn2* IKO mice. Data are expressed as the mean \pm SEM ($n = 3$ mice per group, as indicated).

2. Additional references would bolster the manuscript, both to support claims like (line 90) '2A (p-EIF2A), a marker of amino acid deficiency', which may not be obvious for people outside the field, and to add details regarding methods, like the rationale for using threefold concentrations of amino acids or specific doses and timepoints for GCN2 inhibitor studies.

Our response:

Thank you for your suggestion. We have added additional references to 'p-EIF2A, a marker of amino acid deficiency' and 'three-fold concentration of amino acid' in our revised manuscripts (page 5) respectively. Actually, the references related to specific doses and timepoints for GCN2 inhibitors have been added in the Results (page 7) and Method section (page 14) in our original manuscripts.

3. The manuscript relates its findings to understanding and treating COVID-19. Discussing the results in relation to potential human studies outside the current pandemic would be beneficial for rising the manuscript's significance.

Our response:

Thank you for your suggestion. Reportedly, ACE2 protein was widely expressed in many tissues and with high levels occurring in IECs especially (Hikmet *et al.*, *Mol. Syst. Biol.* 2020; Li *et al.*, *Infect Dis Poverty.* 2020), indicating its important role in intestine physiology. Recent human studies reported that ACE2 levels were elevated in colonic ulcerative colitis (UC) compared with non-inflammatory bowel disease (IBD) controls

(Potdar *et al.*, *Gastroenterology*. 2021). Therefore, in addition to treating COVID-19, we speculated that intervening amino acid levels or inhibiting GCN2 in colon might also provide a strategy for treating colonic UC.

This information has been added to Discussions (page 11) in our revised manuscripts.

4. Description of figures 4 and S7 in the main text does not entirely match the panels of the figures.

Our response:

Thank you for pointing out this issue. We have corrected related description of Figure 4 and S7 in our revised manuscripts (page 8 and 9).

5. AXL (figure S3) is not referenced in the main text as such; adding this gene name in line 95 could be helpful for the reader.

Our response:

Thank you for your suggestion. We have added AXL, the abbreviation of tyrosine-protein kinase receptor UFO, in our revised manuscripts (page 5).

6. Figure S5A has a different Y-scale than all the other bar plots, with 100% mark not labeled.

Our response:

Thank you for your suggestion. We have labeled the Y-scale bar plots with 0, 50%, 100%, and 150%, which was replaced in Figure S8A in our revised manuscripts.

7. Abbreviations section is missing GCN2.

Our response:

Thank you for pointing out this issue. We have added the abbreviation of GCN2 in the Abbreviations section in our revised manuscripts (page 19).

Reviewers' comments:

Reviewer #1 (Remarks to the Author):

The authors have addressed all the issues I raised in the previous round of review. I recommend acceptance for publication.

Reviewer #2 (Remarks to the Author):

My concerns have been addressed and the manuscript was also modified thoroughly.

Reviewer #3 (Remarks to the Author):

The authors responded to the majority of the reviewer's questions and their work helped to fill in important gaps in the manuscript. However, the manuscript could be further improved by addressing the following issues:

1. The RNA-seq data is still unavailable for quality assessment. The BioProject accession number provided (PRJNA808827) could not be found in the database. Attempts to find the datasets in GEO, SRA and BioProject databases using relevant keywords and author names were unsuccessful.

2. The goal of the request for full Western blot membranes was to validate molecular weights of the reported bands and specificity of the antibodies. That the authors provide extended blot images, however they are still not the full membranes, and so:

a) molecular weights cannot be verified and off-target proteins or alternative isoforms (such as dACE2 at ~50kDa) are not visible if present (please see figure S9B and the ATF4 antibody for the best example of an unspecific binding and an uncertain molecular weight)

b) assuming that the unedited blots contain repeats of the same experiment, this shows that Western blot approach can be unreliable at times (for example the unedited blot for figure 2E, where the left-hand experiment does not indicate any GCN2iB effect on ACE2)

The authors are encouraged again to present the full membrane images for the blots used in the figures, including the entire molecular marker ladders. Additionally, the authors could add a paragraph in the Western blot or Statistical Analysis methods section indicating the amount of replicates analyzed and making it clear that the Western blot images are representative of the data.

3. Validation of the IEC-specific knockout mice is appreciated, but not entirely satisfactory. There are significant outliers, especially in the lung and the kidney, displaying three to four-fold difference in protein expression in the GCN2 KO mice. The difference is clear when compared to the intestine in Figure 2D, but RT-qPCR or immunofluorescence, rather than Western blot, may serve better to visualize it.

Point-by-point response to the Reviewers' comments:

Reviewer #1 (Remarks to the Author):

The authors have addressed all the issues I raised in the previous round of review. I recommend acceptance for publication.

Our response:

Thank you very much for your careful reading and good suggestions.

Reviewer #2 (Remarks to the Author):

My concerns have been addressed and the manuscript was also modified thoroughly.

Our response:

Thank you very much for your careful reading and good suggestions.

Reviewer #3 (Remarks to the Author):

The authors responded to the majority of the reviewer's questions and their work helped to fill in important gaps in the manuscript. However, the manuscript could be further improved by addressing the following issues:

1. The RNA-seq data is still unavailable for quality assessment. The BioProject accession number provided (PRJNA808827) could not be found in the database. Attempts to find the datasets in GEO, SRA and BioProject databases using relevant keywords and author names were unsuccessful.

Our response:

We are sorry for this confusion, since we set a delayed release time for the data. We had modified the settings so that the RNA-seq data (PRJNA808827) had been released and could be viewed at website: <https://www.ncbi.nlm.nih.gov/bioproject/PRJNA808827/>.

2. The goal of the request for full Western blot membranes was to validate molecular weights of the reported bands and specificity of the antibodies. That the authors provide extended blot images, however they are still not the full membranes, and so:

a) molecular weights cannot be verified and off-target proteins or alternative isoforms (such as dACE2 at ~50kDa) are not visible if present (please see figure S9B and the ATF4 antibody for the best example of an unspecific binding and an uncertain molecular weight)

b) assuming that the unedited blots contain repeats of the same experiment, this shows that Western blot approach can be unreliable at times (for example the unedited blot for figure 2E, where the left-hand experiment does not indicate any GCN2iB effect on ACE2)

The authors are encouraged again to present the full membrane images for the blots used in the figures, including the entire molecular marker ladders. Additionally, the authors could add a paragraph in the Western blot or Statistical Analysis methods section indicating the amount of replicates analyzed and making it clear that the Western blot images are representative of the data.

Our response:

Thank you very much for your careful reading and good suggestions.

a) To confirm the molecular weights of the reported ACE2 band and specificity of the ACE2 antibody, we performed the ACE2 protein analysis by full membrane. For example, we repeated the analysis of ACE2 expression under GCN2 inhibitor (GCN2iB) stimulation in CCD841 cells by western blot. As shown in Figure S1A, we found that multiple bands were displayed on the full membrane, but only two bands (~110 kDa and ~90 kDa) were obviously observed. Moreover, only the band at approximately 110 kDa was markedly decreased by GCN2iB stimulation. We further validated the truth of the ~110 kDa band by using human *ACE2* siRNA (the target sequence was: 5'-CCATCTACAGTACTGGAAA-3'; Zhang R, et al., 2009, PMID: 19592460). As shown in Figure S1B, after transfected negative control siRNAs (siNC) or human *ACE2* siRNAs (si*ACE2*) into CCD841 cells for 48 h, the *ACE2* mRNA and protein expression at ~110 kDa band were markedly decreased, but the protein expression unchanged at ~90 kDa band, suggesting that the ~90 kDa was the unspecific band of ACE2 antibody in CCD841 cells.

In fact, the antibody used in present study was Rabbit mAb from Abclone (A4612). The validated data from the company as shown in Figure S2, the specific bands of ACE2 protein were 100~120 kDa, which was consistent with our data. Moreover, this antibody was cited by many publications (Bai L, et al., *Cell research* 2021, PMID: 33603116; Xie F, et al., *Adv Mater* 2021, PMID: 34665481). Particularly, Xie F, et al. (2021) confirmed that the molecular weight of the full length human ACE2 protein was ~110 kDa (Figure S3), which was also consistent with our data. Together, although ACE2 have alternative isoforms and unspecific bands, we believed that the ~110 kDa band was specific and selected for ACE2 expression analysis in present study.

Figure S1. Validation of ACE2 bands and antibody specificity. (A) CCD841 cells were incubated with indicated concentration GCN2iB (HY-112654) for 48 h, then ACE2 protein levels were analyzed by western blot. The molecular marker ladders from Thermo 26616. (B) and (C) CCD841 cells were transfected with negative control siRNA (siNC) or siRNAs targeting at human ACE2 (siACE2) for 48 h. ACE2 mRNA and protein levels were analyzed by RT-qPCR and western blot, respectively.

Figure S2. Validation of ACE2 Rabbit mAb (A4612) by western blot. This photo from website: <https://abclonal.com.cn/catalog/A4612>

Figure S3. Bacterially purified ACE2-GFP proteins were analyzed by SDS-PAGE and detected by Coomassie blue staining (From Xie F, et al., Adv Mater 2021, PMID: 34665481).

b) We have repeated the figure S9B and get more specific ATF4 band than the previous data (Figure S4). We have modified the figure S9B in the revised Supplementary Figures.

Figure S4. ATF4 protein expression. CCD841 cells were transfected with control small interfering RNAs (*siNC*) or small interfering RNAs targeting at human ATF4 (*siATF4*). After 12 h, cells were incubated with either control (+ GLN) or glutamine starvation (- GLN) culture medium for 48 h. ATF4 protein levels were analyzed by western blotting (4 replicates per group).

c) We agree with the reviewer's suggestion. We have added the descriptive information about the amount of replicates analyzed in western blot into methods section (page 18) in the revised manuscript. In fact, about the GCN2iB effect on ACE2, we have 4 replicates in the experiments. According to another membrane (Figure S5), the dose of 5 μ M GCN2iB significantly decreased ACE2 expression. These results were consistent with the repeated data shown in Figure S1A.

Figure S5. Regulation of ACE2 expression by GCN2 inhibitor. CCD841 cells were incubated with indicated concentration GCN2iB (HY-112654) for 48 h, then ACE2 protein levels were analyzed by western blot.

3. Validation of the IEC-specific knockout mice is appreciated, but not entirely satisfactory. There are significant outliers, especially in the lung and the kidney, displaying three to four-fold difference in protein expression in the GCN2 KO mice. The difference is clear when compared to the intestine in Figure 2D, but RT-qPCR or immunofluorescence, rather than Western blot, may serve better to visualize it.

Our response:

Thank you for your suggestion. To answer this question, we detected *Gcn2* mRNA expression in lung, stomach, and renal tissue of *Gcn2*-floxed or *Gcn2* IKO mice by RT-qPCR. We found that the mRNA levels of *Gcn2* in these tissues had no difference between the two groups (Figure S6), suggesting that villin cre used in our work had no off-target effects.

This information has been added to Results (page 6) and Supplementary information (Supplementary Figure 7A) in our revised manuscript.

Figure S6. *Gcn2* mRNA expression in lung, stomach, and kidney of intestinal epithelial cell-specific *Gcn2* deletion (*Gcn2* IKO) mice. *Gcn2* mRNA levels were analyzed by RT-qPCR. Studies were conducted using 10-week-old male *Gcn2*-floxed (Con) or *Gcn2* IKO mice. Data are expressed as the mean \pm SEM ($n = 3$ mice per group).